# Shear-Wave Elastography Gradient Analysis of Newly Diagnosed Breast Tumours: A Critical Analysis

**DOI:** 10.3390/diagnostics14151657

**Published:** 2024-07-31

**Authors:** Johannes Deeg, Michael Swoboda, Daniel Egle, Verena Wieser, Afschin Soleiman, Valentin Ladenhauf, Malik Galijasevic, Birgit Amort, Leonhard Gruber

**Affiliations:** 1Department of Radiology, Medical University Innsbruck, Anichstraße 35, 6020 Innsbruck, Austria; johannes.deeg@i-med.ac.at (J.D.); valentin.ladenhauf@i-med.ac.at (V.L.); malik.galijasevic@i-med.ac.at (M.G.); b.amort@tirol-kliniken.at (B.A.); leonhard.gruber@i-med.ac.at (L.G.); 2Department of Obstetrics and Gynaecology, Medical University Innsbruck, Anichstraße 35, 6020 Innsbruck, Austria; daniel.egle@tirol-kliniken.at (D.E.); verena.wieser@tirol-kliniken.at (V.W.); 3Institute for Pathology, INNPath, University Hospital Tirol Kliniken, Anichstraße 35, 6020 Innsbruck, Austria; afschin.soleiman@innpath.at

**Keywords:** shear-wave elastography, breast tumours, SWI gradient

## Abstract

Background: A better understanding of the peritumoral stroma changes due to tumour invasion using non-invasive diagnostic methods may improve the differentiation between benign and malignant breast lesions. This study aimed to assess the correlation between breast lesion differentiation and intra- and peritumoral shear-wave elastography (SWE) gradients. Methods: A total of 135 patients with newly diagnosed breast lesions were included. Intratumoral, subsurface, and three consecutive peritumoral SWE value measurements (with three repetitions) were performed. Intratumoral, interface, and peritumoral gradients (Gradient 1 and Gradient 2) were calculated using averaged SWE values. Statistical analysis included descriptive statistics and an ordinary one-way ANOVA to compare overall and individual gradients among Breast Imaging-Reporting and Data System (BI-RADS) 2, 3, and 5 groups. Results: Malignant tumours showed higher average SWE velocity values at the tumour centre (BI-RADS 2/3: 4.1 ± 1.8 m/s vs. BI-RADS 5: 4.9 ± 2.0 m/s, *p* = 0.04) and the first peritumoral area (BI-RADS 2/3: 3.4 ± 1.8 m/s vs. BI-RADS 5: 4.3 ± 1.8 m/s, *p* = 0.003). No significant difference was found between intratumoral gradients (0.03 ± 0.32 m/s vs. 0.0 ± 0.28 m/s; *p* > 0.999) or gradients across the tumour–tissue interface (−0.17 ± 0.18 m/s vs. −0.13 ± 0.35 m/s; *p* = 0.202). However, the first peritumoral gradient (−0.16 ± 0.24 m/s vs. −0.35 ± 0.31 m/s; *p* < 0.0001) and the second peritumoral gradient (−0.11 ± 0.18 m/s vs. −0.22 ± 0.28 m/s; *p* = 0.037) were significantly steeper in malignant tumours. The AUC was best for *PTG1* (0.7358) and *PTG2* (0.7039). A threshold value for peritumoral *SWI PT1* above 3.76 m/s and for *PTG1* below −0.238 m/s·mm^−1^ indicated malignancy in 90.6% of cases. Conclusions: Evaluating the peritumoral SWE gradient may improve the diagnostic pre-test probability, as malignant tumours showed a significantly steeper curve of the elasticity values in the peritumoral stroma compared to the linear regression with a relatively flat curve of benign lesions.

## 1. Introduction

Breast cancer is the most common malignancy in women at an incidence of up to 92 per 100,000 in highly developed countries, which keeps rising due to an increase in life expectancy [1]. Fortunately, up to 80% of early stage, non-metastatic cases can be cured. Advanced breast cancer, i.e., distant metastasis, is still considered incurable [2], underlining the importance of early detection and treatment.

To facilitate early diagnosis of breast cancer, most developed nations have implemented breast cancer screening programmes with a recommended biennial invitation of women. Such programmes have led to a relative reduction in mortality of up to 20% [3]. Participants undergo standardised mammography and in case of higher breast density, i.e., American College of Radiology (ACR) C or D, additional ultrasound [4]. Diagnosis is usually made based on mammography findings followed by ultrasound and ultrasound-guided biopsy. Over the last several years, there has been diversification into a more multiparametric approach in breast ultrasound beyond conventional B-mode and Doppler sonography. Shear-wave elastography (SWE) quantitatively assesses tissue stiffness by measuring lateral ultrasound propagation speed via Doppler methods [5,6].

In the past years, several studies have shown that malignant breast tumours exhibit higher tissue stiffness with associated higher “shear-wave” velocities compared to benign lesions [7]. Occasionally, some malignant lesions appear “soft” with SWE, although they are very stiff in reality; this is often caused by a poor signal-to-noise ratio. [8]. Besides some false negative results in SWE, there are some malignant lesions such as high-grade ductal carcinoma in situ (DCIS) and papillary cancers with lower elastography values, which make them prone to being missed in elastography [7]. Recent studies have shown that pathogenic mutations in Breast Cancer Gene (BRCA) and non-BRCA genes seem to exhibit benign imaging findings in US patients compared to mutation-negative patients [9].

While most studies have focused on tumoral properties, peritumoral reaction, on the other hand, is a common hallmark of malignant tumours and can include various, most likely immune-mediated, changes to peritumoral tissue based on desmoplastic reaction and tumour cell infiltration [10]. Previous studies have shown that peritumoral tissue can exhibit increased stiffness around malignant lesions [11] and that the evaluation of these changes may provide additional diagnostic information for the evaluation of unclear breast lesions [12].

The aim of the study was to assess a possible correlation between the differentiation of breast lesions and intra- and peritumoral SWE gradients.

## 2. Materials and Methods

### 2.1. Ethics Committee Approval

Study approval for this retrospective study was granted by the Ethical Review Board (ERB proposal 1314/2023).

### 2.2. Study Participants Screening and Inclusion

All women having undergone an ultrasound-guided biopsy of an unknown breast lesion from 1 June 2020 to 31 May 2021 were retrospectively screened. Table 1 provides an overview of the inclusion and exclusion criteria. Among 143 women retrospectively screened, 8 cases had to be excluded due to insufficient ultrasound documentation (*n* = 7) or inconclusive histology (*n* = 1) (Figure 1). Accordingly, 135 women could be included in the study for further analysis.

### 2.3. Ultrasound Equipment, Shear-Wave Elastography Measurements, and Gradient Calculations

All women in our second-level centre undergo a standardised breast ultrasound examination if referred for (1) routine or screening mammography in case of breast density ACR C or D or in case of (2) a suspicious finding upon mammography; (3) referral due to a suspicious finding upon clinical examination; or (4) referral from an extramural radiology practice due to suspicious imaging findings. Examinations are performed on an Acuson Sequoia, Acuson S2000 or S3000 Evolution US scanner with a 18L6 or VF13-5 high-resolution probe (Siemens Healthineers; Erlangen, Germany) encompassing a systematic scanning of the axillary and breast regions. Focal lesions are documented in two perpendicular planes to determine relation to surrounding tissue, size, conventional B-mode, and Doppler properties.

SWE measurements were performed following international guidelines including minimal precompression and avoidance of oblique compression [6]. SWE measurements were performed using 18 or 13.5 MHz linear array transducers. The region of interest (ROI) size was chosen to include the whole lesion and at least 1.5 cm of peritumoral tissue on one or two sides adjacent to the tumour.

If the tumour was larger than the maximum ROI box in dual view mode (5 × 4 cm), then intratumoral and peritumoral measurements were conducted separately. If tumours were too small for intratumoral ROI placement (approximately 5 mm), then only peritumoral measurements were conducted.

After identifying the tumour centre in dual-view B-mode, ROIs were placed at the same tissue depth in the horizontal plane. The placements were as follows: (a) within the tumour centre (ITC), (b) directly beneath the tumour surface (ITS), and (c) peritoumorally at approximately 2 mm (PT1), 5 mm (PT2), and 8 mm (PT3) from the tumour surface (with exact distances measured). This is illustrated in Figure 2 and Figure 3. For each location, three respective measurements were taken, both vertically and horizontally aligned. After averaging SWE results for each region, SWE gradients were calculated between (1) tumour centre and tumour periphery (“intratumoral gradient”, *ITG_SWE_*), (2) interface gradient between tumour periphery and peritumoral area (“interface gradient”, *ING_SWE_*), (3) between the first (PT1) and second (PT2) peritumoral measurement area (“peritumoral gradient 1”, *PTG1_SWE_*), and (4) between the second (PT2) and third (PT3) peritumoral measurement area (“peritumoral gradient 2”, *PTG2_SWE_*).

After finishing the standard ultrasound examination including SWE measurements, all patients underwent ultrasound-guided biopsy following histological assessment.

### 2.4. Core-Needle Biopsy Procedure

To achieve definitive diagnoses, breast biopsies are carried out following international guidelines [13]. A detailed explanation is given to each patient and a signed informed consent is obtained afterwards. After thorough skin disinfection and intracutaneous and ultrasound-guided perilesional application of a local anaesthetic (Mepivacaine hydrochloride 1%), a small skin incision is made. Then a 12 G or 14 G core-needle biopsy system (HistoCore Automatic Biopsy System, BIP GmbH, Türkenfeld, Germany) is used to acquire five tissue specimens under constant ultrasound-guidance. Specimens are then embedded in a 5% formalin solution for further staining and analysis.

### 2.5. Histologic Evaluation

Following WHO guidelines [14], histological evaluation is carried out by specialised gynaecological pathologists, including histopathological type, nuclear grade, peritumoral spread, and immunohistochemical (IHC) results (oestrogen receptor (ER), progesterone receptor (PR), HER-2, and Ki-67 status). If available, results from full resection specimen analysis were used, if not, those from core-needle samples. If any form of neoadjuvant therapy had been administered since initial diagnosis, the histopathological core-needle biopsy reports were used instead.

Histopathological data were gathered from the hospital’s clinical information system, KIS PowerChart (Cerner, North Kansas City, MO, USA).

### 2.6. Statistical Analysis

Statistical analysis was carried out using the statistics program GraphPad Prism version 10.2.3 (GraphPad Software, La Jolla, CA, USA) and SPSS^®^ version 29 (International Business Machines Corporation, Armonk, NY, USA). Descriptive statistical analysis was performed after classification of cases. After descriptive statistical analysis, group characteristics were compared using an ordinary one-way ANOVA with a Holm-Sidak correction or Kruskal-Wallis test with Dunn’s post-test (in case of non-Gaussian distribution). Univariate analyses of binary and nominal variables were performed using cross-tabulations. Correction for multiple testing was applied where necessary. Receiver operating characteristic curves were generated using SWE values and gradients, and corresponding area-under-the-curve (AUC) values were calculated. Threshold values for continuous variables were selected following Youden’s J. *p*-values < 0.05 were considered statistically significant, and *p*-values < 0.1 signified a trend towards significance.

## 3. Results

### 3.1. Patient Demographics and Lesion Characteristics

Overall, 143 female patients were retrospectively screened. After exclusion of 8 cases due to insufficient ultrasound documentation (*n* = 7) or inconclusive histology (*n* = 1), 135 patients were included. The average patient age was 59.6 ± 13.9 years (range 24.2 to 84.7 years).

The average tumour size was 13.9 ± 11.8 mm (median 11 mm, range 1.6 to 105.0 mm). Among all tumours, 23.7% (*n* = 32) were of benign differentiation, 4.4% (*n* = 6) were of intermediate differentiation, and 71.9% (*n* = 97) were of malignant differentiation. Within malignant cases, grade 1 tumours accounted for 13.4% (*n* = 13), grade 2 for 59.8% (*n* = 58), and grade 3 for 26.8% (*n* = 26).

The most common benign diagnoses were fibroadenomas (48.3% [*n* = 14]), followed by glandular tissue/adenosis (27.6% [*n* = 8]) and fibrous-cystic mastopathy (20.7% [*n* = 6]). The most common malignant diagnoses were invasive carcinomas of the breast of no special type (NST) (72.3% [*n* = 68]) and ductal carcinoma in situ (16.0% [*n* = 15]) (Figure 4).

### 3.2. SWE Velocities

Overall, malignant tumours showed higher average SWE velocity values for the tumour centre ITC (BI-RADS 2/3: 4.1 ± 1.8 m/s vs. BI-RADS 5: 4.9 ± 2.0 m/s, *p* = 0.04) and the first peritumoral area PT1 (BI-RADS 2/3: 3.4 ± 1.8 m/s vs. BI-RADS 5: 4.3 ± 1.8 m/s, *p* = 0.003). While the other regions did not differ significantly between BI-RADS 2/3 and BI-RADS 5 lesions, there was an overall significant influence of histology (*p* < 0.0001) and area (*p* < 0.0001) on SWE velocity (Figure 5).

### 3.3. Gradient Analysis

While there was no significant difference between intratumoral gradients *ITG_SWE_* (0.03 ± 0.32 m/s vs. 0.0 ± 0.28 m/s; *p* > 0.999) or gradients across the tumour–tissue interface *ING_SWE_* (−0.17 ± 0.18 m/s vs. −0.13 ± 0.35 m/s; *p* = 0.202) (Figure 6a), the first peritumoral gradient *PTG1_SWE_* (−0.16 ± 0.24 m/s vs. −0.35 ± 0.31 m/s; *p* < 0.0001) and second peritumoral gradient *PTG2_SWE_* (−0.11 ± 0.18 m/s vs. −0.22 ± 0.28 m/s; *p* = 0.037) were significantly steeper in malignant tumours (Figure 6b).

When comparing BI-RADS 2/3 lesions, non-invasive and invasive BI-RADS 5 lesions DCIS and invasive carcinomas, only invasive BI-RADS 5 tumours showed a significantly steeper peritumoral gradient *PTG1_SWE_* and *PTG2_SWE_* (Figure 7).

### 3.4. Diagnostic Utility

When comparing SWE and gradient ROC AUC values, PTG1 had the highest AUC of 0.74. Overall, ROC curves of peritumoral gradient-based analysis showed higher AUC values compared to velocity-based measurements. The same did not hold true for intratumoral assessment, though, where gradients did not distinguish malignant from other tumours, whereas velocity measurements did at a moderate AUC of 0.625. For a graphical comparison, please refer to Figure 8. The two highest-scoring variables *PT1* and *PTG1* were selected to generate a scoring system. Threshold values for peritumoral *SWE PT1* were 3.76 m/s and for *PTG1*, −0.238 m/s·mm^−1^ (short explanation: −0.238 m/s·mm^−1^ means that for every millimetre moved in the specified direction, the SWE speed decreases by 0.238 m/s). Cases with both above-threshold *PT1* and below-threshold *PTG1* were malignant in 90.6% (Table 2).

## 4. Discussion

We report the first results of a retrospective study that aimed to explore the diagnostic utility of intra- and peritumoral SWE gradients with potential implications for ultrasound-based BI-RADS classification of newly diagnosed breast tumours.

Several previous studies have evaluated the benefits of SWE measurements in the differentiation of benign and malignant breast lesions [7,8]. In many of those previous studies, the elasticity values of a breast lesion were measured by placing multiple ROIs in the stiffest part or the centre of the respective lesions. Instead of focusing on the centre of the breast lesions, we paid special attention to the peritumoral stroma (PS), as this zone may represent a crucial interface between the tumour and the host’s native surrounding tissue.

We could reproduce previous findings on benign lesions having lower SWE values across all points measured, i.e., intratumorally and peritoumorally. Furthermore, benign lesions demonstrated, on average, a linear and relatively flat decrease in SWE values from within the tumour to the peripheral peritumoral stroma. Compared to these findings, malignant lesions exhibited significantly steeper peritumoral gradients, stemming from higher peritumoral SWE values. Contrary to our expectations, intratumoral gradients did not differ between benign and malignant tumours.

Those changes to the peritumoral stroma constitute one of the hallmark characteristics of malignant tumours. On the one hand, the peritumoral stiffness is a result of abnormal low-elasticity collagen associated with cancer fibroblasts, enhanced PI3 kinase activity, and direct infiltration of cancer cells [15,16,17]. On the other hand, the stiffness is influenced by an increase in angiogenesis and microvessel density [18], and by immune-related cellular infiltration and physiologic alterations reflecting increased metabolic demands [19]. Those changes form a more or less spatially organised “ecosystem”, aptly proposed by Sofopoulus et al. [20].

The steeper decrease pattern of the peritumoral SWE values for malignant lesions is in accordance with recent studies from Park et al. [11], yet our study even showed a linear decrease pattern for benign lesion and not a flat curve as in the aforementioned study. A pathophysiological reason behind this finding is probably simply rooted in the local compression of the mammary tissue, even around a benign lesion without any direct infiltration or host reaction [21,22].

A direct comparison to the work by Yang et al. [23] is much more difficult because the authors only measured the peritumoral changes up to 3 mm, which in our case still corresponds to the peritumoral measurement point P1, and therefore our measurement points P2 and P3 have no correlate. Still, the results of Yang et al. also illustrate a certain decrease in stiffness in the first 3 mm around a breast lesion for both benign and malignant lesions.

This may be especially important for DCIS, as it is constituting a non-obligate precursor of invasive ductal carcinoma [24]. As described in the literature, if left untreated, approximately 12% of the patients develop an invasive form, even if only focally invasive in most cases [25]. Currently, the treatment of choice is surgical removal and radiotherapy to reduce risk of recurrence [26]; however, there is a controversy over overtreatment of DCIS patients, as not all progress to invasive disease. Hypothetically, the evaluation of the peritumoral changes may add a new diagnostic tool to find exactly those DCIS patients who exhibit some form of invasiveness. The reason for those changes is the disruption of the myoepithelium and the basement membrane [27], which then facilitates the tumour invasion into the stroma. Given the limited number of cases of DCIS in our study, our analysis was primarily observational in nature. Nonetheless, a notable observation emerged: low-grade DCIS cases typically exhibited no to minimal peritumoral changes in contrast to high-grade DCIS.

Therefore, understanding the changing pattern of tissue stiffness across tumour borders may have additional diagnostic value, and may change the treatment of early-stage breast cancer, especially in older patients, for whom no major operations can be carried out because of their general health condition, and who may therefore undergo thermal ablation therapy like cryo- [28] or radiofrequency ablation [29]. Here, these peritumoral areas with steep gradients may be considered part of a necessary “safe zone” in minimal invasive treatment options, in our opinion. Moreover, the evaluation of peritumoral tissue and gradients presents another avenue for exploration in future studies aimed at leveraging deep learning computer algorithms, which are gaining prominence in the field of breast cancer diagnostics [30]. Moreover, the determination of peritumoral gradient measurements is expected to undergo considerable simplification and enhanced objectivity in the future through advancements in radiomics analyses.

Besides any treatment changes, better evaluations of a breast lesion due to better imaging features would also be valuable information for the interpreting pathologist, as differentiation between certain grades of breast cancer can be histologically difficult depending on the quality of tissue specimens and can vary among pathologists [12,31,32].

The comparison of the tumour centre and the peritumoral tissue regarding the tissue stiffness showed no significant differences, as some other studies described higher SWE values in the peritumoral tissue compared to the tumour centre [21,33]. Even though recent studies have yielded varied results regarding overall intratumoral SWE cut-off values between benign and malignant lesions, our findings were well within this range, similar to the recent studies from Golatta et al. [8] with a cut off of 3.7 m/s, Berg et al. [21] at 5.2 m/s, Chang et al. [34] at 5.2 m/s, and Tozaki et al. [35] at 3.6 m/s. Nevertheless, the evaluation of the central tumour stiffness was not the main aim of this study. Interestingly, there was no difference in intratumoral gradients between benign and malignant tumours. One explanation may be that the tumours in our cohort were on average not larger than 14 mm and therefore too small to develop significant central necrosis. In a cohort with larger invasive carcinomas, the presence of central necrosis may be more common, and thus intratumoral gradients may in fact be significantly steeper from the tumour centre to the periphery.

The gradient SWE measurement also may have an advantage over the isolated measurements of the inner stiffness of the tumour, as some “stiff” malignant tumours may have a poor SNR, which can lead to a “soft” appearance; this is due to the fact that the shear wave is able to propagate in hard lesions, but due to a poor SNR of the shear-wave motion, it can be difficult to obtain a valid shear-wave speed estimation [36]. To avoid those misinterpretations, the investigator can look at the provided additional quality map, where those areas are often easier to identify as artefactual [37].

Overall, gradient analysis may be more robust in certain aspects compared to a SWE-values based approach. The gradient measurement may not be influenced by artefacts in the tumour centre, but there are others. In the case of a tumour very close to the skin or the chest wall, for example, “compression” artefacts often occur and can thereby lead to incorrectly high stiffness values. As only relative changes are assessed, results are not as susceptible to overall tissue changes such as after radiotherapy or a higher degree of fibroglandular tissue [38]. Furthermore, user-based biases such as suboptimal precompression [6,39] may also be cancelled out to some degree by a gradient analysis.

### Limitations

Several limitations need to be reported. First, this study is retrospective in nature and single-centre. Like other shear-wave studies, this study is also subject to restrictions due to the physical conditions of this examination method [5,40].

Furthermore, the case number can be considered only moderate and the distribution of malignant tumours favouring no special type (NST) may introduce some bias towards other types of invasive carcinomas. Moreover, it is important to acknowledge that our grouping of individual tumour types under the BI-RADS 5 category may introduce potential statistical distortions. This is because various tumour types can exhibit significant differences in presentation. Even though the number of histopathologically verified benign lesions is lower than that of the malignant lesions, this is a reflection of the clinical routine.

SWE measurements are generally considered to be user-independent and reproducible [8]. However, challenges may arise when positioning regions of interest (ROIs) for tumours with poorly defined edges, where ROI placement may vary depending on orientation. To ensure objectivity and enhance measurement reliability, we positioned ROIs in areas with minimal artefact superimposition in the horizontal orientation, aiming for consistency regardless of tumour orientation or boundary clarity. Despite these efforts, potential inaccuracies due to imprecise ROI placement cannot be entirely ruled out. Additionally, inter- and intra-observer variability is a well-known and recognised issue in sonography in general.

Another limitation of this study pertains to our decision to place ROIs exclusively on one side of the tumour. Although circular path measurements theoretically offer advantages, we encountered significant challenges with vertically oriented measurements due to artefacts induced by precompression. Additionally, while considering extending measurements to both left and right along the horizontal axis, practical constraints led us to opt against this approach. Instead, we chose to position ROIs solely on sides displaying the most notable peritumoral changes. Nevertheless, future advancements, particularly in computer-aided measurements and ROI placement, hold promise for simplifying this process.

## 5. Conclusions

An improved understanding of the peritumoral stroma changes due to tumour invasion by non-invasive and reliable diagnostic methods may improve the identification of malignant tumours and reduce the number of unnecessary biopsies. A SWE gradient-based analysis may constitute a valuable additional tool, as 90.6% of tumours with a peritumoral SWI above 3.76 m/s and peritumoral gradient below −0.238 m/s·mm^−1^ were malignant.

## Figures and Tables

**Figure 1 diagnostics-14-01657-f001:**
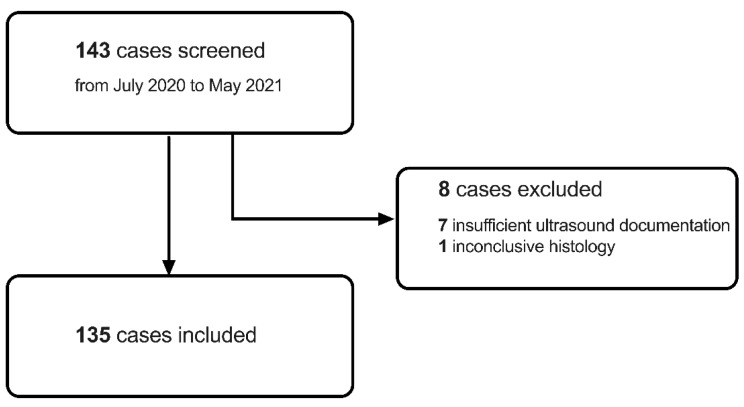
Overview of participant screening, exclusion, and inclusion.

**Figure 2 diagnostics-14-01657-f002:**
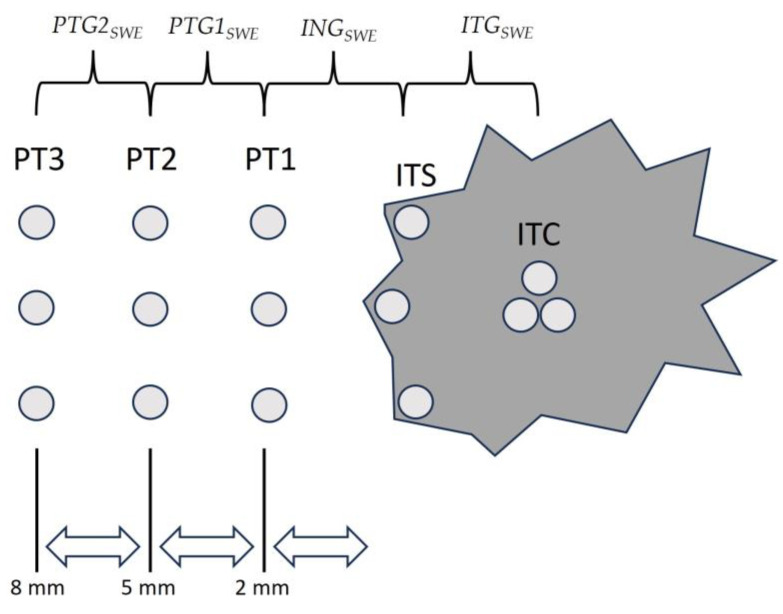
Illustration of intra- and peritumoral shear-wave elastography (SWE) region-of-interest (ROI) placement (ITC: tumour centre, ITS: tumour surface, PT1: peritumoral 1, PT2: peritumoral 2, PT3: peritumoral 3, *ITG_SWE_*: intratumoral gradient, *ING_SWE_*: interface gradient, *PTG1_SWE_*: peritumoral gradient 1, *PTG2_SWE_*: peritumoral gradient 2).

**Figure 3 diagnostics-14-01657-f003:**
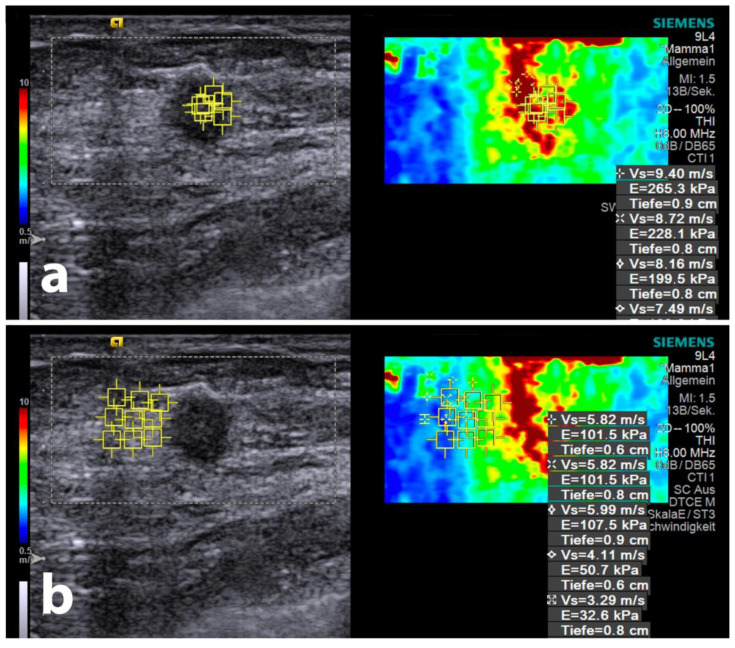
Ultrasound image of intra- (**a**) and peritumoral (**b**) shear-wave elastography (SWE) region-of-interest (ROI) placement.

**Figure 4 diagnostics-14-01657-f004:**
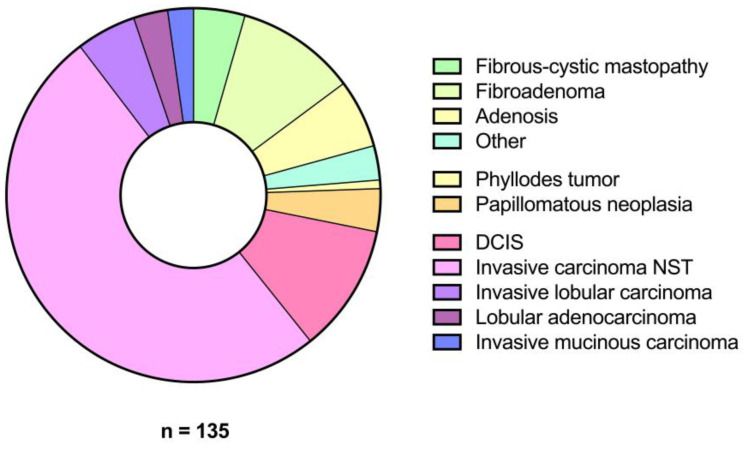
Distribution of histological tumour subtypes.

**Figure 5 diagnostics-14-01657-f005:**
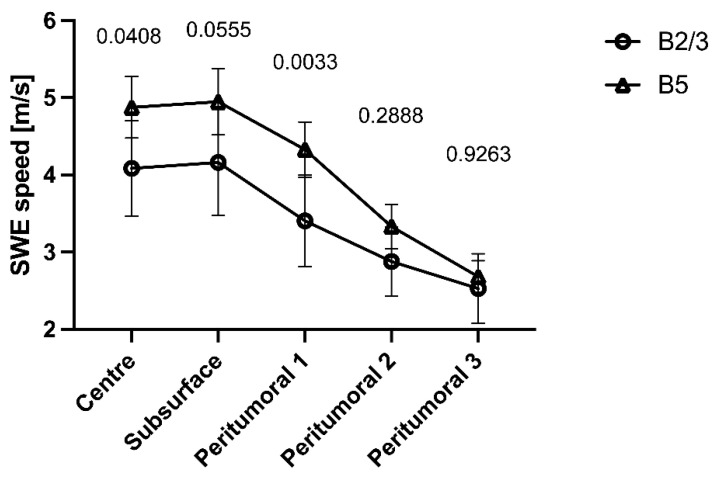
Comparison of shear-wave elastography velocities measured across the tumour centre, subsurface region, and peritumoral regions 1–3 for B2/3 (*n* = 38) and B5 (*n* = 97) lesions.

**Figure 6 diagnostics-14-01657-f006:**
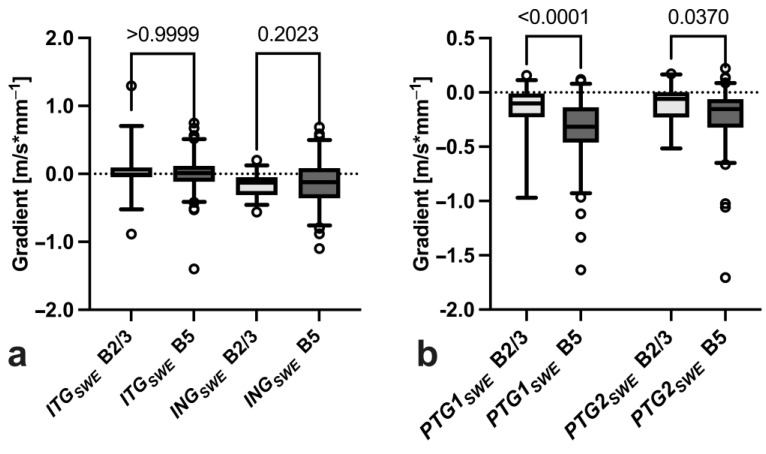
Comparison of intratumoral (**a**) and peritumoral SWE gradients (**b**) between BI-RADS 2/3 (*n* = 38) and BI-RADS 5 (*n* = 97) breast tumours.

**Figure 7 diagnostics-14-01657-f007:**
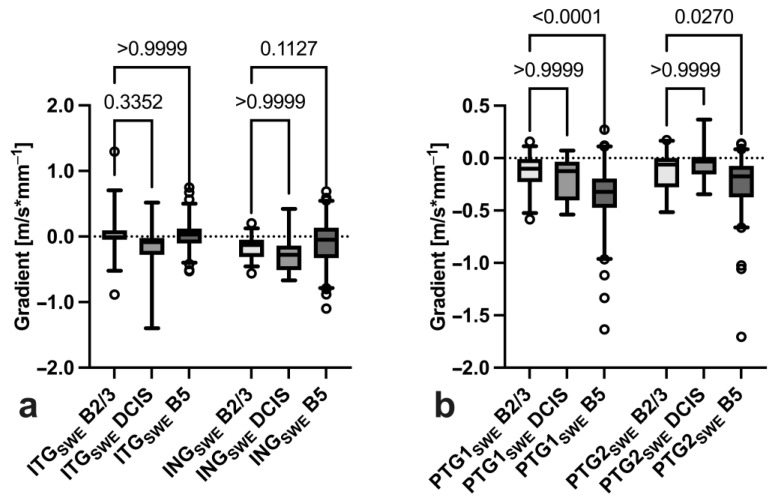
Comparison of intratumoral (**a**) and peritumoral SWE gradient slopes (**b**) between BI-RADS 2/3 (*n* = 38), DCIS (*n* = 15) and invasive BI-RADS 5 (*n* = 82) breast tumours.

**Figure 8 diagnostics-14-01657-f008:**
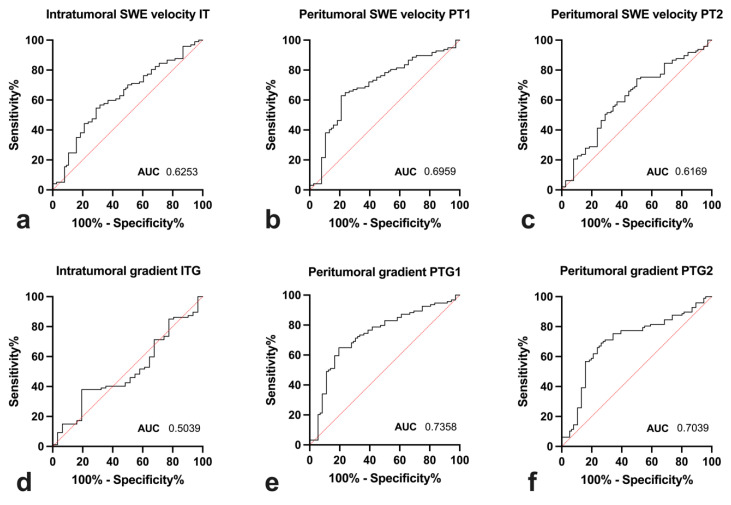
Receiver operating characteristics curve for (**a**–**c**) average shear-wave elastography measurements and (**d**–**f**) shear-wave elastography gradients.

**Table 1 diagnostics-14-01657-t001:** Inclusion and exclusion criteria.

Inclusion Criteria	Exclusion Criteria
Ultrasound-guided biopsy of newly diagnosed breast tumour	Patient age under 18 years
Sufficient ultrasound documentation including B-mode and shear-wave elastography imaging	Prior history of ipsilateral breast cancer
Available histological diagnosis	Insufficient ultrasound documentation
	Inconclusive histology

**Table 2 diagnostics-14-01657-t002:** Frequency of malignant differentiation grouped by peritumoral SWE speed and gradient.

		Peritumoral SWE Speed PT1	
		≤3.76 m/s	>3.76 m/s
Peritumoral gradient PTG1	≥−0.238 m/s·mm^−1^	46.0%	81.3%
	<−0.238 m/s·mm^−1^	76.5%	90.6%

## Data Availability

Data will be made available on request to authors.

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
