# Peer review of "Shear-Wave Elastography Gradient Analysis of Newly Diagnosed Breast Tumours: A Critical Analysis"

_diagnostics, 2024, doi:10.3390/diagnostics14151657_

Round 1

Reviewer 1 Report

Comments and Suggestions for Authors

Summary: The reviewer understood that the aim of this study was to investigate a possible correlation between the differentiation of breast lesions and intra- and  

peri-tumoral shear-wave elastography gradients. The results showed that benign lesions showed a linear regression of the elasticity values from the centre of a lesion to the peripheral peritumoral stroma with a relative flat curve, whereas malignant lesions showed a significant steep, and there was significant different patterns between benign and malignant lesion. 

Strengths: Its an interesting topic and has potential application value.

Weakness: The study design was problematic, the results were doubtful, and the conclusion was not reliable.

The authors said this study was a retrospective study. However, the measurement of shear-wave velocity must be performed in real-time elastography, so it can not be completed retrospectively.

Agreement of intra/inter observers must be considered. ICC of the measurement of shear-wave velocity must be tested.

Comments on the Quality of English Language

Quality of English Language Is Excellent. The word stroma used in the manuscript is not appropriate.

Author Response

Dear reviewer. Thank you for the kind words.

Major comments:

Comments  1: The study design was problematic, the results were doubtful, and the conclusion was not reliable.

Response 1: Thank you for your feedback on our study. We greatly appreciate your insights, which have have tried to incorporate.

We appreciate your criticism of the study design, yet hope to have addressed it below.

We have tried to address the issues you raised as follows:

  • Study Design: see Point 2
  • Results: We have further refined our analysis and presentation of the results to provide a clearer and more comprehensive understanding of our findings.
  • Conclusion: The conclusion has been thoroughly revised to better reflect the results, ensuring that it is now more reliable and accurately represents the implications of our study.

We hope that these revisions meet your expectations and adequately address your concerns. Thank you once again for your constructive critique.

Comments 2: The authors said this study was a retrospective study. However, the measurement of shear-wave velocity must be performed in real-time elastography, so it can not be completed retrospectively.

Response 2: Thank you for your feedback on our study. We appreciate the opportunity to clarify the points you raised.

While it is true that shear-wave velocity measurements need to be performed in real-time, we would like to highlight that in our department, these measurements have been integrated into our standard routine procedures. Since shear-wave elastography is non-invasive and does not harm the patient, we routinely include these measurements during patient examinations without requiring additional time or effort.

Therefore, although our study is retrospective in nature, the shear-wave velocity data was collected during routine clinical practice, allowing us to include it in our retrospective analysis.

We hope this explanation clarifies the methodology used in our study. Thank you again for your valuable input.

Comments 3: Agreement of intra/inter observers must be considered. ICC of the measurement of shear-wave velocity must be tested.

Response 3: Thank you for your valuable feedback regarding the intra- and inter-observer agreement and the necessity to test the ICC of shear-wave velocity measurements.

We acknowledge the importance of these considerations and have addressed the known issues of intra- and inter-observer variability in shear-wave elastography in the "Limitations" section of our manuscript. We have elaborated on these challenges and introduced gradient measurements to move away from fixed shear-wave velocity values, which can for example be affected by the compression applied by the examiner. For instance, SWI values can vary between examiners due to differing levels of probe compression on the tissue. Arguably, gradient measurements remain relatively unaffected by this variability as we have strictly adhered to standards set among other by the EFSUMB. Furthermore, even if compression might alter the absolute values at individual measurement points, the gradient between the two points remains consistent, thus providing a more robust assessment. This new approach aims to reduce intra- and inter-observer variability.

Nevertheless, we must acknowledge that some degree of inaccuracy between different examiners remains unavoidable. This is a well-recognized issue in ultrasonography, and unfortunately, it cannot be completely eliminated.

Thank you once again for your insightful comments, which have significantly helped us improve our study

Reviewer 2 Report

Comments and Suggestions for Authors

The authors used the peri-tumoral shear wave elastograpy gradient to classify the benign and malignant breast lesions in a retrospective study. The proposed gradient was compared with BI-RADS in a statistical analysis. They found that the SWE-gradient is a good feature in recognize the patterns between benign and malignant lesions, which may provide a sensitive tool in the clinical diagnosis.

It’s unclear whether the peri-tumoral shear wave elastography gradient is determine along a certain direction or averaged value around the tumor and how to determine the center or boundary of tumor.

Line 44 delete calculations

Line 58 whats ACR?

Line 64 change in the last years to in the past years

Line 67 grammar error

Figure 2 the dimensions or criteria of determining PT1, PT2, PT3 should be included

Line 176 how to determine a tumor accurately as small as 1 mm as the resolution of sonography is only about 1 mm?

Table 2 the unit of SWE speed gradient seems strange

Comments on the Quality of English Language

overall, it is good

Author Response

Dear reviewer. Thank you for the kind words.

Major comments:

Comments 1: It’s unclear whether the peri-tumoral shear wave elastography gradient is determine along a certain direction or averaged value around the tumor and how to determine the center or boundary of tumor.

Response 1: As the reviewer correctly pointed out, our initial description of the ROI placement was somewhat imprecise. To clarify this, we have made the necessary revisions in the main manuscript. We identified the center of the tumor using the dual view in B-mode and conducted subsequent measurements from this reference point. (see line 142-148). While the central measurement points were grouped, the subsurface and peritumoral measurement ROIs were  placed vertically to the right or left of the tumor and at the same depth to avoid vertical axis values distortion due to the pressure applied by the ultrasound probe. (see therefore line 383-391 in the limitations section).

Minor comments:

Comments 1: Line 44 delete “calculations”

Response 1: We appreciate the tip and have updated the main manuscript accordingly

Comments 2: Line 58 what’s “ACR”?

Response 2: We are grateful for your input. Unfortunately, we initially omitted to spell the abbreviation ACR. ACR is an abbreviation for American College of Radiology. The recommended changes have been made in the main manuscript. (line 81)

Comments 3: Line 64 change “in the last years” to “in the past years

Response 3: Thank you for the advice. The main manuscript has been revised to reflect these changes.

Comments 4: Line 67 grammar error

Response 4: Thank you for the tip, we have updated the main manuscript accordingly.

Comments 5: Figure 2 the dimensions or criteria of determining PT1, PT2, PT3 should be included

Response 5: Thank you for the suggestion. We have incorporated the changes into the main manuscript. (see Figure 2)

Comments 6: Line 176 how to determine a tumor accurately as small as 1 mm as the resolution of sonography is only about 1 mm? 

Response 6: The theoretical maximum spatial resolution of the 18-MHz system we used is 0.04 mm (f = 18x106/sec, c = 343x103 mm, pulse n = 4). Still, after going through out data, the smallest tumour we found was indeed 1.6 mm in size. This mistake was corrected. In such small tumours no intratumoral measurements were conducted. This was added to the methods sections.

“…were conducted separately. If tumours were too small for intratumoral ROI placement (approximately 5 mm), then only peritumoral measurements were conducted.” (Line 139-141)

Comments 7: Table 2 the unit of SWE speed gradient seems strange

Response 7: The unit of gradient measurement may initially look a bit confusing, which is why we have made it a little clearer here. We hope that it is now a little more understandable. (see line 254-256) “"m/s*mm⁻¹"means that the speed (in meters per second) changes by a certain amount for each millimeter of distance. For example, if you have a gradient of 2 m/s*mm⁻¹, it means that for every millimeter you move in the specified direction, the speed increases (or decreases) by 2 meters per second.

Reviewer 3 Report

Comments and Suggestions for Authors

The abstract needs quantification. Why only p values and gradient value are chosen for the tumor classification? The research gap is to be analyzed. The limitations are to be included. Better representation of ROC is required than specificity Vs Sensitivity.  The discussion on ROC may be improved. The conclusion may be modified. The experimental setup and screening for the patients are good. The need for this case study may be included with demographic study.

Author Response

Dear reviewer. Thank you for your input.

Major comments:

Comments 1: The abstract needs quantification. Why only p values and gradient value are chosen for the tumor classification? The research gap is to be analyzed. The limitations are to be included. Better representation of ROC is required than specificity Vs Sensitivity.  The discussion on ROC may be improved. The conclusion may be modified. The experimental setup and screening for the patients are good. The need for this case study may be included with demographic study.

Response 1: As the reviewer correctly noted, our abstract required some revisions. Initially, we aimed to adhere to the specified word/character limit, which led to the shortened version. According to the journal's regulations, including the limitations in the abstract is not mandatory and would exceed the specified word/character limit. Therefore, we have omitted them from the abstract but discussed them in detail in the limitations section. We have now revised the abstract and conclusion in the main document to incorporate your suggestions.

Furthermore, more details were added to the results section regarding the ROC analysis.

“Abstract:

Background: A better understanding of the peritumoral stroma changes due to tumor invasion using non-invasive diagnostic methods may improve the differentiation between benign and malignant breast lesions. This study aimed to assess the correlation between breast lesion differentiation and intra- and peri-tumoral shear-wave elastography (SWE) gradients. Methods: A total of 135 patients with newly diagnosed breast lesions were included. Intratumoral, subsurface, and three consecutive peritumoral SWE value measurements (with three repetitions) were performed. Intratumoral, interface, and peritumoral gradients (Gradient 1 and Gradient 2) were calculated using averaged SWE values. Statistical analysis included descriptive statistics and an ordinary one-way ANOVA to compare overall and individual gradients among Breast Imaging-Reporting and Data System (BI-RADS) 2, 3, and 5 groups. Results: Malignant tumors showed higher average SWE velocity values at the tumor center (BI-RADS 2/3: 4.1 ± 1.8 m/s vs. BI-RADS 5: 4.9 ± 2.0 m/s, p = 0.04) and the first peritumoral area (BI-RADS 2/3: 3.4 ± 1.8 m/s vs. BI-RADS 5: 4.3 ± 1.8 m/s, p = 0.003). No significant difference was found between intratumoral gradients (0.03 ± 0.32 m/s vs. 0.0 ± 0.28 m/s; p > 0.999) or gradients across the tumor-tissue interface (-0.17 ± 0.18 m/s vs. -0.13 ± 0.35 m/s; p = 0.202). However, the first peritumoral gradient (-0.16 ± 0.24 m/s vs. -0.35 ± 0.31 m/s; p < 0.0001) and the second peri-tumoral gradient (-0.11 ± 0.18 m/s vs. -0.22 ± 0.28 m/s; p = 0.037) were significantly steeper in ma-lignant tumors. The AUC was best for PTG1 (0.7358) and PTG2 (0.7039). A threshold value for peritumoral SWI PT1 above 3.76 m/s and for PTG1 below -0.238 m/s*mm⁻¹ indicated malignancy in 90.6% of cases. Conclusion: Evaluating the peritumoral SWE gradient may improve the diagnostic pre-test probability, as malignant tumors showed a significantly steeper curve of the elasticity values in the peritumoral stroma compared to the linear regression with a relatively flat curve of benign lesions.” (Line 26-47)

“5. Conclusion:

An improved understanding of the peritumoral stroma changes due to tumour inva-sion by non-invasive and reliable diagnostic methods may improve the identification of malignant tumours and reduce the number of unnecessary biopsies. A SWE gradient-based analysis may constitute a valuable additional tool, as tumours with a peritu-moral SWI above 3.76 m/s and peritumoral gradient below -0.238 m/s*mm⁻¹ were malignant in 90,6%.” (Line 392-398)

“Results:

3.4. Diagnostic Utility

When comparing SWE and gradient ROC AUC values, PTG1 had the highest AUC of 0.74. Overall, ROC curves of peritumoral gradient-based analysis showed higher AUC values compared to velocity-based measurements. The same did not hold true for intratumoral assessment, though, where gradients did not discern malignant from other tumours, whereas velocity measurements did at a moderate AUC of 0.625. For a graphical comparison, please refer to Figure 8. The two” (Line 246-252)

Round 2

Reviewer 1 Report

Comments and Suggestions for Authors

Strengths: It’s an interesting topic and has potential application value.

Weakness: The study design was problematic, the results were doubtful, and the conclusion was not reliable.

The authors said this study was a retrospective study. However, the measurement of shear-wave velocity must be performed in real-time elastography, so it can not be completed retrospectively. 

Agreement of intra/inter observers must be considered. ICC of the measurement of shear-wave velocity must be tested.

The authors responses to the reviewers previous comments were just some explanation, the issues remained. 

The reviewer admit that some sonographers measured the shear-wave velocity using the similar methods described in this manuscript, but it's not competent, they could not do well and acquire a reliable results.

The reviewer firmly believes that an appropriate well designed study is must for such study.

Reviewer 3 Report

Comments and Suggestions for Authors

All the corrections are included in the paper. Hence, There is no need for further review of the article.